# Engagement in water governance action situations in the Lake Champlain Basin

**Patrick Bitterman**[1]*, **Christopher Koliba**[2]

**1** School of Global Integrative Studies, University of Nebraska-Lincoln, Lincoln, NE, United States of America, **2** School of Public Affairs and Administration, University of Kansas, Lawrence, KS, United States of America

\* patrick.bitterman@unl.edu

## Abstract

Water quality governance encompasses multiple "wicked" interacting problems that manifest within social-ecological systems. Concerned governments, institutions, and actors concerned with addressing these issues must wrestle with complex systems that span time, space, and scale. This complexity of connected systems requires the participation of multiple actors across political boundaries, problem areas, and hydrologic domains. In Lake Champlain (US), frequent cyanobacteria blooms negatively affect property values, recreational activities, and public infrastructure, in addition to their impacts on the aquatic ecosystem. Through a survey of actors working on water quality in the Lake Champlain Basin, we analyze how actor participation in structured issue forums creates a network of connected action situations across multiple spatial scales and problem domains. We apply exponential random graph models to quantify the effects of scale, issues, and homophily on actor participation in these forums. Our findings show that actors tend to favor participating in similarly scoped forums at their spatial scale, that actors are less likely to participate in structured forums focused on agriculture, and that actors participate independently of others with whom they have pre-existing collaborative relationships. Further, we find that in the case of the Lake Champlain Basin, actor participation in issues related to water quality is misaligned with modeled sources of nutrient pollution. This study demonstrates that the design and maintenance of water quality action situations play an important role in attracting the participation of actors working collaboratively to address wicked social-ecological problems. Further, linking current and potential configurations of governance networks to social-ecological outcomes can aid in the effective and efficient achievement of management objectives.

## 1. Introduction

Cyanobacterial harmful algal blooms (cyanoHABs) are common in water bodies across the United States. CyanoHABs often produce toxins harmful to the aquatic environment and humans, but they also negatively affect property values, recreational activities, and public infrastructure [1]. The occurrence and prevalence of cyanoHABs in freshwater bodies result from

**Data Availability Statement:** The data have been made publicly available on the University of Nebraska-Lincoln Data Repository, and can be found here: https://doi.org/10.32873/unl.dr.20230224

**Funding:** PB and CK were supported by the National Science Foundation under VT EPSCoR Grant No. NSF OIA 1556770, Lake Champlain Basin Resilience to Extreme Events (BREE) (https://www.nsf.gov/awardsearch/showAward?AWD_ID=1556770). The funders had no role in study design, data collection and analysis, decision to publish, or preparation of the manuscript.

**Competing interests:** The authors have declared that no competing interests exist.

complex interactions among nutrient runoff from agriculture, atmospheric deposition, waste-water, stormwater, industrial outputs, legacy nutrients, and the impacts of climate change [2–7]. While the importance of interactions among climate and nutrients is becoming increasingly clear [5, 8], the relative contribution of these factors varies substantially by local spatial context within the affected waterbody and its watershed. Given the breadth of the causes and impacts of cyanoHABs, associated water governance institutions concerned with reducing the problem(s) must wrestle with complex systems that span time, space, and scale. This multiplicity of connected complex systems further means that addressing the issues requires the participation of multiple actors across political boundaries, problem domains (e.g., engineering, public health, economics), and hydrologic domains. Accordingly, understanding how water governance actors interact in formal and informal settings can improve our understanding of how adaptive management, policy interventions, and learning can affect cyanoHABs and their impacts on human health and environmental quality. Further, linking current and potential configurations of governance networks to social-ecological outcomes can aid in the effective and efficient achievement of management objectives [9].

In Lake Champlain, cyanoHABs frequently occur in the late summer months along the shoreline and in the shallow bays of the northeast portion of the lake. Blooms in Lake Champlain result from interactions among nutrient pollution (primarily phosphorus), climatic conditions, and benthic phosphorus [8]. However, the key driver of these cyanoHABs is the nutrient pollution that comes from multiple land uses/land covers across the Lake Champlain Basin (LCB), including forestry, urban development, and agriculture [10–12]. To address these issues, the United States Environmental Protection Agency (USEPA) issued a total maximum daily load (TMDL) regulation for phosphorus pollution in the LCB in 2016. The state of Vermont has also enacted multiple pieces of legislation to control land use and fund clean water projects to reduce bloom-contributing nutrient pollution [13, 14]. However, significant management challenges substantially hamper the achievement of clean water goals in Lake Champlain.

First, integrated assessment modeling has shown that as Lake Champlain warms due to climate change, additional legacy nutrients trapped in lake sediment will be released, further fueling blooms even if surface nutrient runoff was dramatically reduced [8]. The same study showed that due to this interaction between benthic phosphorus and warming, the codified targets set by the TMDL are insufficient for reducing blooms. Second, the reductions in nutrient pollution that would be required to achieve clean water goals cannot be met solely under the jurisdiction of existing federal and state laws. This is because most private land use–especially agricultural activities–are a major contributor to the problem and are not subject to these regulations. In response, the Vermont Agency of Natural Resources (VTANR) has increased its focus on what are termed "non-regulatory projects", or voluntary projects funded by the state, but not mandated by federal law [14]. Achieving water quality goals through a mix of voluntary and mandated programs requires a system of collaborative governance–one where public and private stakeholders work formally and informally to make management decisions [15, 16]. Accordingly, education, engagement, and outreach to private, non-profit, and public actors (e.g., organizations, institutions, and agencies) concerned with LCB water quality are increasingly important to achieving changes to the water quality regime. The water quality issues in the LCB are not unique to the region, and there are many similar TMDLs across the US [17]. However, the multiplicity of engaged actors across the basin, as well as the varied approaches employed by VTANR and other state agencies, have created model conditions to study collaborative governance in practice.

There are hundreds of actors engaged in water quality and quantity issues across the LCB. Many of these actors engage with each other directly through the sharing of information,

resources, or technical assistance [18–20]. However, other interactions are mediated by actor participation in forums, groups, or across networks. While the structure and function of these assemblages can vary greatly, they can be classified under the umbrella term of an *action situation* [21]. In the vocabulary of Ostrom, action situations are a unit of analysis where actors interact with each other and with their environment, and do so in the context of institutional rules to produce potential outcomes. Action situations are linked [22, 23] and may include social, ecological, and social-ecological interactions [24]. Understanding which actors participate in which forums and what does (or does not) spur collaborative participation can help identify gaps in the collaborative governance system that might limit successful adaptive response.

Participation in, and engagement with, other actors and environmental issues are commonly modeled using social network and social-ecological network analyses. For example, social-ecological network analysis is generally concerned with the fit between two networks–the social network that manages the environment and environmental function(s) in question (e.g., surface hydrology, wildfire, fisheries) [25–29]. Within the social network, we can describe edges as being between two actors or between an actor and an action situation it participates in. It follows that we can then identify commonalities in action situations and in the *participation in* action situations to build a network of action situations themselves [22, 30]. This is foundational to the *ecology of games* framework [31–33], which describes how understanding the multiple policy games that operate simultaneously in a social-ecological system can lead to better adaptive management.

Collaboration among actors can take many forms, including the exchange of information, coordination of projects, or technical assistance, among other modes. Actors build trust and mutually-beneficial relationships through collaboration [34, 35] and are more likely to participate in forums that reduce transaction costs of collaboration [36]. Cross-scale coordination is important for the governance of problems spanning multiple scales [37, 38], though cross-scale interactions are commonly dominated by more powerful and well-resourced organizations [39]. In the LCB, state agencies such as VTANR fill this role, though both their centrality and the structure of the network are highly dependent on the mode of coordination [18]. Further, the polycentric nature of water governance in the state has generated strong regional forums (e.g., Tactical Basin Planning Committees, Clean Water Advisory Councils) that are geographically bound and serve prescribed collaborative functions.

There is evidence that forums attract actors with similar issues, political beliefs, or policy preferences, as transaction costs to collaboration are lower in these settings [40–42]. However, the strength (and in some cases, direction) of this homophilic effect depends on the particular context, and it may be secondary to issues of trust or access to resources [43, 44]. In some social-ecological contexts, collaboration among actors is more likely when they jointly participate in particular forums [45–47]. This collaborative closure effect is dependent on the scale of activity, but can also reduce transaction costs and strengthen relationships among actors [45]. We expect that the presence of established watershed-scale forums in the LCB, the relatively small size of the system, and the general openness to participation in Vermont would result in higher rates of joint participation in the various forums, which we explore below.

Our analysis uses survey data to measure how participation by actors across multiple action situations connects forums and issues in the basin, and how that participation is dependent on the scale and scope of both the forums and actors. We expect forums with complementary and connected functionality to be more closely linked by joint participants. We also measure the degree to which actors in the LCB water quality governance system participate in an array of action situations across multiple scales and related domains. Specifically, we test the influence of spatial scale, focal issues of action situations, scale homophily, and issue homophily on actor

participation. Here, our expectation is that actors will generally exhibit homophily with respect to scale and issues of concern. That is, actors will favor forums where their issues align with the collective purpose and cover the same jurisdictional scale across which they operate. We also anticipate actors will participate in forums their existing partners also engage in, which we test below. Through this analysis, we identify potential gaps in actor participation in the collective, collaborative governance of water quality issues across the LCB. For example, the institutional characteristics that increase participation or the factors (e.g., scale and issue homophily) that constrain adaptation and require additional attention from managers.

## 2. Methods

### 2.1 Data sources

Data were collected via a web-based survey of actors (e.g., organizations, agencies, institutions) engaged in water quality or quantity issues in the Vermont or Québec portions of the LCB. The survey was active from July–December, 2019. The initial set of possible respondents was seeded from previous surveys in the basin [19, 20] and was supplemented by expert knowledge, document analysis (e.g., meeting minutes), and internet searches. Our lists were validated by expert staff at VTANR. Subjects were contacted via email and asked to respond on behalf of the actor they represent. Up to two follow-up emails were sent to non-respondents. We received responses from 88 of the 203 (43.3%) actors contacted. The response rate was affected by our inclusion of many private firms and small organizations without permanent staff. To increase confidence that our sample adequately measures the governance network, VTANR staff validated that our sample captures nearly all major actors in the LCB water governance system.

The survey asked about actor activities, including participation in resource management issues (e.g., agriculture, stormwater, forestry), measures of accountability and oversight (e.g., from judicial rulings, professional codes of conduct), the services they provide (e.g., grants, technical assistance), and other actors with whom the actor collaborates. Most relevant to this analysis, we also asked which forums or policy response efforts the respondent's organization engages in (see Table 1).

**Table 1. Eight action situations related to water quality management in the Lake Champlain Basin.** Scale and issues coded by authors.

| Action situation name | Scale | Issues of concern | Description |
|---|---|---|---|
| Municipal stormwater planning | watershed | development, stormwater | Includes local commissions, boards, and other planning groups related to stormwater management at the municipal scale |
| Clean Water Network | LCB | wastewater, forestry, agriculture, development, stormwater | A statewide network of networks with more than 100 member organizations dedicated to creating a culture of clean water across the state [48]. |
| Tactical Basin Planning Committees | watershed | wastewater, forestry, river corridors, agriculture, development, stormwater | Committees of experts and stakeholders that engage in planning clean water activities at the "tactical basin" scale (approximates HUC-8 watersheds) [49] |
| Green Infrastructure Collaborative | LCB | development, stormwater | Partnership between Lake Champlain Sea Grant and Vermont Department of Environmental Conservation. Promotes green stormwater infrastructure and low impact development [50] |
| Clean Water Advisory Committees | watershed | wastewater, forestry, river corridors, agriculture, development, stormwater | Oversee regional policy development and activities related to achieving Lake Champlain TMDL |
| The Watershed Alliance | LCB | river corridors, agriculture, stormwater | Educational program managed by Lake Champlain Sea Grant [51] |
| Legislative Committees on Natural Resources | LCB | wastewater, forestry, river corridors, agriculture, stormwater | State legislative committees responsible for developing policy related to natural resources |
| Legislative Committees on Agriculture | LCB | forestry, agriculture | State legislative committees responsible for developing policy related to agriculture and forestry |

We further coded responses to assign a jurisdictional scale to each actor and action situation. Assigning a single spatial scale to an actor or forum can be challenging, as some actors operate across multiple scales, as do the issues at the focus of some action situations. In Vermont, there is an ongoing shift in the management of non-regulatory projects from state agencies to regional Clean Water Service Providers (CWSPs) [14]. To align our analysis with this change in policy, we simplify our coding of spatial scale to two levels. The "watershed scale" includes actors or action situations that operate within 2 or fewer HUC-8 scale watersheds. The group includes actors such as resource conservation districts, regional planning commissions, and municipal boards. "LCB scale" actors and action situations work across the entire Lake Champlain Basin, and in many cases, much more broadly. This group encompasses state and national agencies, international commissions, and private firms (e.g., engineering consultants). While this assigned scale is a simplification of the governance system, it more closely matches the state-to-local scale transition. Finally, we also coded the primary issues of concern for each action situation to align with the survey questions regarding actor function. These lists of action situations emerged from a qualitative assessment and document analysis of the region's water quality policy domain. While these forums and issues are not exhaustive, they possess qualities common to regional freshwater water quality governance systems. Our final list includes eight water quality action situations of interest (Table 1).

## 2.2 Analysis

Network data structures were generated from survey data using the R package *tidygraph* [52] and visualized using the R package *ggraph* [53] in the R statistical programming language [54]. In the network, nodes correspond to actors or action situations, while edges represent relationships connecting the nodes (Fig 1). We began with a network containing two types of nodes–actors and action situations. We calculated the degree of each node in the network, which corresponds to the number of incoming or outgoing edges from a node. The overall degree distribution of the network is exponential, with many actors having few edge connections to other nodes. Programs and offices within VTANR and other state agencies are highly central, suggesting a governance network focused more on state-led activities [18].

The network plotted in Fig 1 contains two types of edges–those between two collaborating actors, and those representing an actor's participation in a structured water governance action situation. To analyze the connections among action situations themselves, we collapsed the network to only include nodes corresponding to action situations (Fig 2, described below). In this graph, the edges are weighted by the number of actors that jointly participate in both action situations. Thus, we approximate how connected two action situations are via their participants.

To measure the influence of scale and focal issues on actor participation, we utilized exponential random graph models (ERGMs). An ERGM assumes the measured network is one realization of many potential networks. Using Markov Chain Monte Carlo (MCMC) simulations, the model produces similar networks to the observed network, generating statistics related to parameter estimation (e.g., the effects of scale homophily on the relative likelihood of an edge occurring between two nodes) and overall model fit [55, 56]. We fit two ERGMs using the *STATNET* packages in R [57]. As inputs into these models, we produced a bipartite network retaining only those edges corresponding to actor participation in action situations. Our research questions are concerned with how institutional design (e.g., scale, issue domain) can affect actor participation in collaborative governance action situations. Accordingly, we first created a baseline model to estimate the effects of actor and action situation issues of concern, actor and action situation scale, and issue and scale homophily between actors and action

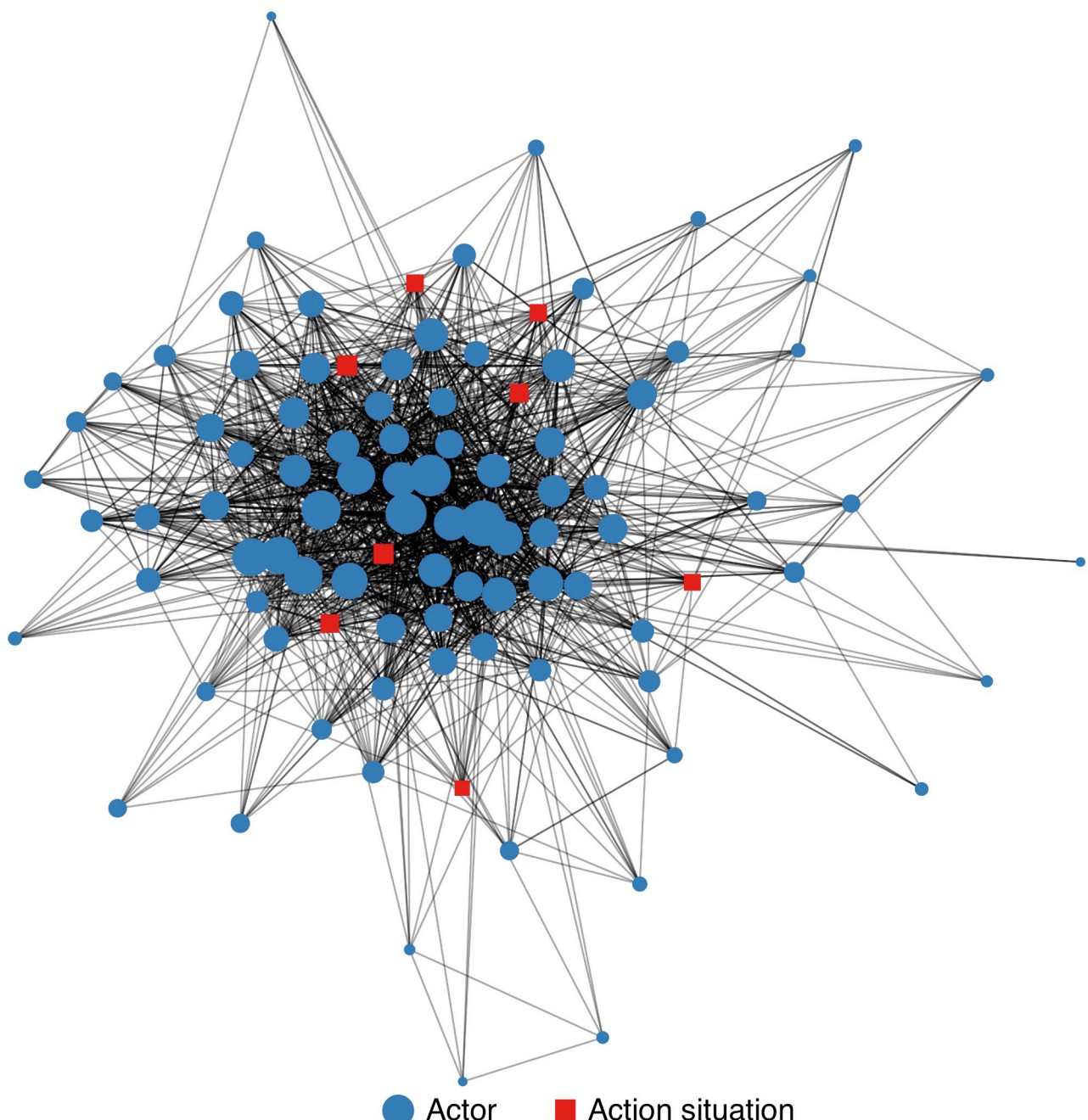

**Fig 1. Network of actors (blue circles) and action situations (red squares) engaged in water quality governance in the LCB.** Nodes are scaled relative to their degree centrality.

situations. We also use a control parameter to account for the geometrically-weighted degree distribution of actors (gwb1degree) and action situations (gwb2degree) to control from structural characteristics of the network. In fitting the model, we began by including all possible factors and simplified the model to exclude non-explanatory terms.

The second ERGM extends the first model to test whether collaboration among two actors led to greater co-participation in the same action situation. There is some evidence in the

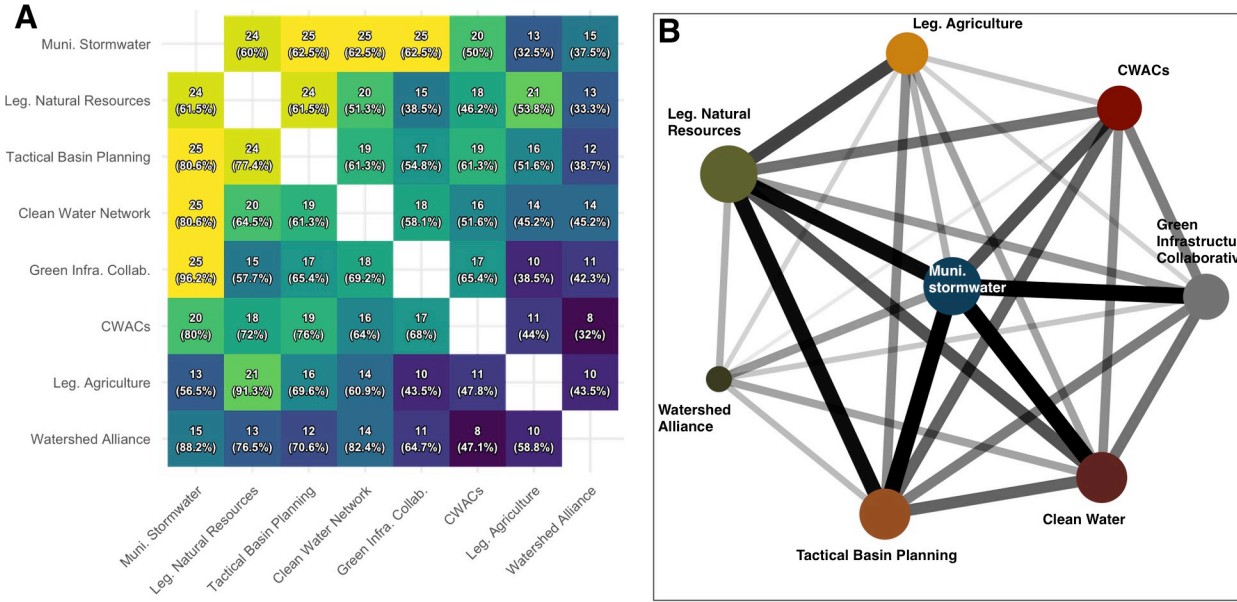

**Fig 2. Connected action situations.** (A) Joint participation in the various action situations. Values correspond to the number of actors jointly participating in action situation pairs. The percentage of actors participating from the action situation listed on the left are shown in parentheses. (B) Action situation relationships plotted as an undirected graph. Node size is scaled to the number of participant). Width and shading of edges are scaled according to co-participation.

literature that, at least for some resource management issues, this collaborative closure predicts participation [45]. For example, that direct coordination between $actor_A$ and $actor_B$ positively affects the likelihood that both will participate in action $situation_Z$. To model these effects, for each actor-action situation edge, we calculated the number of actors connected to the focal actor that also participate in the same action situation. We did so for 5 modes of actor-actor coordination (i.e., information sharing, technical assistance, financial exchange, reporting, and project coordination), and modeled these values using *edgecov* terms. All terms and model diagnostics are described in the S1 File.

## 3. Results

### 3.1 Descriptive analysis

The degree of surveyed participation in water quality action situations is presented in Table 2, ranging from approximately 45% to about 20% of respondents. The action situation garnering

**Table 2. The number of respondents that participate in the water quality action situations in the Lake Champlain Basin.** Percent of total respondents participating included in parentheses.

| Action situation | Participants (% of respondents) |
| --- | --- |
| Municipal stormwater planning | 40 (45.5%) |
| Legislative committees on natural resources | 39 (44.3%) |
| Tactical Basin Planning Committees | 31 (35.2%) |
| Vermont Clean Water Network | 31 (35.2%) |
| Green Infrastructure Collaborative | 26 (29.5%) |
| Clean Water Advisory Councils | 25 (28.4%) |
| Legislative committees on agriculture | 23 (26.1%) |
| The Watershed Alliance | 17 (19.3%) |

**Table 3. Actor engagement in surveyed water quality-related issues and load allocations from the 2016 TMDL.** Issue engagement is non-exclusive, so the percentages sum to more than 100%.

| Surveyed issue | Load source (2016 TMDL) | Pct. of actors engaged | Reductions required to meet 2016 TMDL (by pct.) |
|---|---|---|---|
| Agriculture | Agricultural nonpoint | 62.9% | 53.6% |
| River corridors | River instability | 67.1% | 45.4% |
| Forestry | Forested lands | 55.7% | 18.7% |
| Stormwater | Developed Lands | 80% | 20.9% |
| Development | | 58.6% | |
| Wastewater | Sewage treatment plants | 52.9% | 42.1% |

the most participation is municipal stormwater planning (over 45% of respondents), followed by legislative committees on natural resources (44.3%), Tactical Basin Planning Committees (35.2%), and the Vermont Clean Water Network (35.2%).

In Fig 2, we present joint participation in the various action situations. In Fig 2A, we include the number of actors that jointly participate in both action situations. While the network is undirected, overall counts of participation vary by action situations, so the percentage of participation varies by the baseline. For example, 24 actors participate in both municipal stormwater planning and legislative committees on natural resources. Those 24 actors make up 60% of all actors participating in municipal stormwater, but 61.5% of those participating in natural resources committees with the Vermont state legislature. The axes are sorted such that action situations with greater overall participation are in the top-left corner, while lower participation is in the bottom-right.

In Fig 2B, we plot these relationships as an undirected graph. Node size is scaled to the number of participants in each action situation (as shown in Table 2). The width and shading of edges are scaled according to co-participation in connected action situations. Here, we see tighter links among municipal stormwater planning committees, legislative committees on natural resources, Tactical Basin Planning Committees, the Clean Water Network, and the Green Infrastructure Collaborative. Weaker relationships are found between legislative committees on agriculture and most other nodes, and between Clean Water Advisory Committees and the Watershed Alliance.

In addition to engagement with the above action situations, we calculated actor engagement with related water quality issues. In Table 3, we show this issue engagement alongside the estimated reductions of total phosphorus (TP) required to meet the 2016 Lake Champlain TMDL [58]. The actors we surveyed are most engaged in stormwater, river corridors, and agricultural issues. However, the greatest TP reductions are specified to come from agricultural land uses, followed by river instability and sewage treatment plans. Reductions in developed land use are estimated to be approximately 21%, despite its attraction to LCB actors.

## 3.2 ERGM results

The results of the ERGMs expand upon the descriptive analysis to explain what factors predict actor participation in the LCB action situations. The first model measures the impacts of scale, issues of concern, and scale and issue homophily on the likelihood of an edge from an actor to an action situation. We iteratively reduced the model to exclude many of the non-significant variables and improve model fit. Model results are shown in Table 4.

This first model indicates that actors at the watershed scale (those not operating across the entire LCB) are more likely to engage in structured participation via the included action situations. Similarly, those action situations that also operate at the watershed scale are more likely to attract participants. Unsurprisingly, we also find that scale homophily also predicts greater

**Table 4. ERGM results.** Model 1 contains terms related to spatial scale and focal issues of concern. Model 2 includes all terms from model 1 and adds joint collaboration as explanatory variables.

| Parameter | ERGM 1: Estimate (Std. Error) | ERGM 2: Estimate (Std. Error) |
|---|---|---|
| Actor scale (watershed) | 0.42 (0.16)** | 0.5 (0.23)* |
| Action situation scale (watershed) | 0.75 (0.31)* | 0.31 (0.34)[NS] |
| Scale homophily | 0.76 (0.25)** | 0.65 (0.27)[NS] |
| Action situation issue: agriculture | -0.97 (0.25)*** | -0.8 (0.27)** |
| Action situation issue: development | 0.07 (0.29)[NS] | 0.39 (0.31)[NS] |
| Issue homophily: agriculture | 0.25 (0.22)[NS] | 0.16 (0.25)[NS] |
| Issue homophily: development | 0.81 (0.22)*** | 0.69 (0.25)** |
| Joint collaboration: information sharing | | 0.05 (0.06)[NS] |
| Joint collaboration: financial exchange | | -0.06 (0.09)[NS] |
| Joint collaboration: project coordination | | 0.12 (0.11)[NS] |
| Joint collaboration: reporting | | 0.05 (0.09)[NS] |
| Joint collaboration: technical assistance | | 0 (0.08)[NS] |
| Edges | 10.03 (1.43)*** | 4.98 (1.74)** |
| Actor GW Degree ($\theta_S = 3.2$) | -12.52 (1.62)*** | -7.73 (1.91)*** |
| Action situation GW Degree ($\theta_S = 4.5$) | -2.06 (2.05) | -1.07 (3.83) |
| AIC | 553.6 | 468.7 |
| BIC | 590.5 | 526.2 |

Significance code

*** p-Value < 0.001

** p-Value < 0.01

* p-Value < 0.05

† p-Value < 0.1

[NS] not significant.

participation–LCB-scale actors engage in LCB-scale action situations, and watershed-scale actors participate in watershed-scale action situations. With respect to issues of concern, we find that action situations focused on agriculture are relatively less likely to attract participation, and there is no evidence for homophily in the agricultural sector. Conversely, homophily in development-centric issues does predict greater participation, despite those action situations not attracting greater participation than other issues. Control parameters and model fit diagnostics are described in greater detail in the S1 File.

The second ERGM extends the first model to test the impacts of joint collaboration with other actors on their participation in action situations. Here, we find that having an actor's collaborators participate in an action situation has no significant impact on whether the focal actor participates in that same action situation (Table 4). The inclusion of additional terms, many of them with non-significant effects, alters model fit and the influence of other terms. For example, and despite this model converging with similar diagnostics, the second model does not find the same effects of scale on actor participation as the prior model.

## 4. Discussion

Our results show participation in water quality-related action situations in the LCB spans multiple functions, including local municipal stormwater management, regional planning, and engagement with the state legislature. Previous survey research in the LCB has found that despite the many heterogeneous actors in the water governance system, network centrality and substantial water quality activities are heavily concentrated in a small number of actors,

primarily at the state jurisdictional scale [18]. Our findings extend this work to measure actor engagement via structured and semi-structured action situations across multiple problem domains and spatial scales.

As measured by joint participation, some action situations are much more closely connected than others. For example, we see substantial overlap within the triad of HUC-8 scale Tactical Basin Planning Committees, legislative committees on natural resources, and local stormwater planning. These connections reflect key processes within Vermont's governance system. The state legislature sets rules regarding the mandates and funding for tactical basin planning committees, which attempt to reduce nutrient load through various activities, including reducing local stormwater runoff. The feedback loop closes, as local experts and planners engage with policymakers to provide feedback and influence laws and rulemaking. While our descriptive analysis measures pairwise connections among action situations, future work could extend our analysis to investigate sets of action situations to measure participation and overlap among actors.

The issues our surveyed actors focus on do not entirely align with the agreed-upon reductions required to meet the TMDL requirements. For example, we find an apparent oversubscription to issues on urban land uses, with 80% of actors engaged with stormwater issues and almost 60% in development issues, despite that sector requiring a 21% reduction in load. A similar gap is present in participation in issues related to river corridors. While this may indicate functional scale mismatches between the source of pollution and engagement with the issues [59], we caution against equating engagement with the required effort to reduce loads in any given sector. Some sectors are much easier to regulate than others (e.g., stormwater vs. agriculture), and it is also likely that a 15% reduction in one sector is substantially more difficult than the same reduction in another.

The first ERGM found clear effects of spatial scale and scale homophily on actor participation in action situations. However, the effects of water quality issues and issue homophily on participation are more variable and context-dependent. For example, only two (out of six) issues of concern had significant effects. The positive effects of homophily in development issues may be related to the permitting processes and public hearings required by Vermont law for new developments. Our findings also indicate that action situations concerned with agricultural issues had comparatively lower engagement than other action situations. This supports the idea that homophily is dependent on the particular context of the issue or policy [43, 44].

Our results may reflect how we defined an "actor" in this study as the private and public organizations, institutions, and firms engaged in water quality, and did not include Vermont farmers in our sample frame. However, previous work with Vermont farmers has shown their adoption of nutrient best management practices (BMPs) is largely a function of how much control farmers feel they have over their management practices [60]. Further, in the US, the roles of government and other third parties are limited. Conservation practices and other BMPs are typically funded by voluntary federal programs (e.g., the National Resource Conservation Service), which were outside of the scope of this project. However, the connections among agriculture, natural resource conservation, and the production–and elimination–of ecosystem services across multiple spatial scales is well documented [61–65]. In the LCB, much of the discussion around agricultural practices has centered on payment for ecosystem services (PES) schemes, which are voluntary programs that pay for farmer performance in reducing phosphorus runoff [66, 67]. Accordingly, it is not unexpected to see these apparent tensions reflected as a lack of ties in the networks.

In our second model, we extended the first to include terms related to joint participation by collaborating actors in the same action situations. Previous research has shown there is a greater likelihood of collaboration among actors if they both participate in the same forums

[46] and that linkages within a given scale are more likely than those spanning scale [45]. Our model does not test the direction of causality with respect to collaboration predicting action situation participation or vice versa. However, our results unexpectedly show that collaboration among actors does not impact their participation in the action situations we included. In particular, it is surprising that direct project coordination relationships among actors are unrelated to their participation in related forums, suggesting that the forums are, more than likely, structured to provide strategic direction, advocacy, and information sharing–and less so operational alignments. These findings suggest that collaborative closure in social networks is context-dependent–and is related to the type of collaboration taking place among actors *and* on the function and scale of the forums through which they participate. Many of these forums are focused on policy, advocacy, and formation sharing. Accordingly, it is possible the breadth of activities provided by VTANR across the network has an outsized impact on our analysis as well. Collaborative governance in the LCB, as mediated through structured action situations, may benefit from collaborative engagement in technical assistance and other functional support.

Our analysis shows the participation of actors in action situations is predicated on the spatial scale of both the jurisdiction of actors and the scope of the action situation. Actors that work across the whole of the LCB are more likely to participate in similarly-scoped action situations, as are those actors that operate more locally. Our previous work has also shown that space matters in collaborative governance, in that LCB actors are more likely to participate with actors in nearby watersheds [18]. Place-based research that combines governance analysis and environmental modeling can deepen our understanding of complex, adaptive systems [68]. Our findings extend polycentric governance frameworks that chart how actor participation can connect forums across space and scale [22, 32, 69] by showing how those connections are differentiated by the type of issue and by geographic scale. In Vermont, this may prove particularly relevant, as the aforementioned CWSPs were created to coordinate non-regulatory water quality projects at the HUC-8 scale, thereby shifting the scale of management in the state [14]. As the CWSPs will be intrinsically connected to HUC-8 watersheds, they will represent new place-based foci for polycentric governance across the LCB [70, 71]. Our analysis has identified gaps in water quality forum participation by sector, which may indicate where both VTANR and CWSPs may want to focus engagement efforts.

More generally, cyanoHABs are a water quality issue in many watersheds across the US [1] and around the world, and relevant actors commonly work across and through multiple social-ecological action situations. These forums may be connected in other ways other than actor participation, including the sharing of rules, environmental processes or focal problems, or institutional infrastructure [30]. Greater attention to the construction of these action situations is critical to attract the participation of key actors necessary for collaborative governance to work, and these dynamics should be considered by policymakers and managers as they craft collaborative forums to tackle social-ecological problems. If designed carefully, regionally-bound institutions can improve nutrient pollution and resource efficiency [9]. And, as we have shown in the LCB, these forums are vehicles for engagement, and regional entities such as Vermont CWSPs will need to rely on the creation of relevant action situations to garner the involvement of actors working on water quality problems.

## 5. Conclusion

This work utilized an extensive survey of water quality governance actors to analyze their participation in water quality forums and to derive connections among water quality action situations in the Lake Champlain Basin. Our research emphasizes the complexity of water

governance across the LCB, especially as it pertains to collaboration across spatial scales and issue engagement. Reducing the extent and severity of cyanoHABs in Lake Champlain will require addressing pollution from multiple sources while also coping with legacy nutrients and the impacts of climate change. As we have shown, the many water quality-related action situations in the LCB reach a substantial fraction of water governance actors in the system. However, there are substantial gaps in participation with agricultural issues and agricultural actors. Institutions such as the Vermont CWSPs, as well as similar regional water governance entities across the US, enter complex, multi-scale, multiplex, polycentric water governance systems containing heterogeneous actors, issues, and forums [72]. Researchers and policymakers alike should pay close attention to the role action situations–which are usually purposefully created and maintained by concerned actors–play in addressing wicked social-ecological problems.

## Supporting information

**S1 Fig. Goodness of fit metrics for ERGM #1.**
(TIF)

**S2 Fig. Goodness of fit metrics for ERGM #2.**
(TIF)

**S1 File. Description of exponential random graph models parameters and interpretation.**
(PDF)

## Author Contributions

**Conceptualization:** Patrick Bitterman.

**Data curation:** Patrick Bitterman.

**Formal analysis:** Patrick Bitterman.

**Funding acquisition:** Christopher Koliba.

**Investigation:** Patrick Bitterman, Christopher Koliba.

**Methodology:** Patrick Bitterman.

**Project administration:** Patrick Bitterman, Christopher Koliba.

**Resources:** Christopher Koliba.

**Software:** Patrick Bitterman.

**Supervision:** Patrick Bitterman, Christopher Koliba.

**Visualization:** Patrick Bitterman.

**Writing – original draft:** Patrick Bitterman.

**Writing – review & editing:** Patrick Bitterman, Christopher Koliba.

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
