## [Decision Letter · Decision Letter 0]

6 Dec 2022

PONE-D-22-24459Engagement in Water Governance Action Situations in the Lake Champlain BasinPLOS ONE

Dear Dr. Bitterman,

Thank you for submitting your manuscript to PLOS ONE. After careful consideration, we feel that it has merit but does not fully meet PLOS ONE’s publication criteria as it currently stands. Therefore, we invite you to submit a revised version of the manuscript that addresses the points raised during the review process.

ACADEMIC EDITOR: Thank you for submitting your work to Plos One. Two reviewers have now completed their assessments, and, on the basis of these assessments, I am recommending minor revisions. Both reviewers provide constructive criticisms that do not require new analyses, though additional analysis and clarification may be required in some instances. A area for improvement identified by both reviewers concerns the the generality of the results and how these contribute to a body of theory. Revier1 states: `` I believe the paper would benefit from a stronger explanation of the study’s findings beyond the Lake Champlain Basin study setting." Reviewer 2 notes: ``What are key theoretical puzzles that remain unresolved that you want to answer with regards to the impacts of scale, issues of concern, and scale issue homophily, and joint collaboration on actor participation in the action situations? What are the findings of existing studies, gaps/puzzles that remain, and how does your research address them?" Addressing these critiques will, I believe, improve the reach of the contribution and contribute to the scientific soundness of the manuscript for publication.

We look forward to receiving your revised manuscript.

Kind regards,

Jacob Freeman

Academic Editor

PLOS ONE

Journal Requirements:

Reviewers' comments:

Reviewer's Responses to Questions

**Comments to the Author**

1. Is the manuscript technically sound, and do the data support the conclusions?

Reviewer #1: Yes

Reviewer #2: Yes

2. Has the statistical analysis been performed appropriately and rigorously? 

Reviewer #1: Yes

Reviewer #2: Yes

3. Have the authors made all data underlying the findings in their manuscript fully available?

Reviewer #1: No

Reviewer #2: Yes

4. Is the manuscript presented in an intelligible fashion and written in standard English?

Reviewer #1: Yes

Reviewer #2: Yes

5. Review Comments to the Author

Reviewer #1: This paper evaluates a set of expectations about how actors participate in structured issue forums that address water quality governance challenges in the Lake Champlain Basin, USA. The authors use data from a survey of these actors to estimate network models to evaluate how patterns of participation are shaped by the scales at which actors and forums operate, the focal issues that forums address, and the joint focus of actors and forums on the same scales and issues. As the authors point out, there is need for greater understanding of how actors navigate governance systems that are complex in the sense that they encompass multiple interacting policy issues, and the study’s findings contribute significantly to this understanding. My main comment: I believe the paper would benefit from a stronger explanation of the study’s findings beyond the Lake Champlain Basin study setting. I address this point in more detail below. My other comments are divided into major and minor points.

Major comments:

- Nearly all of discussion focuses on the specific context of water quality governance in the Lake Champlain Basin. I encourage the authors to highlight the broader implications of the study. Likewise, the authors may consider explaining—more explicitly—the importance of the study setting in the introduction (e.g., is this a model system for studying institutional complexity? Are HAB or other water quality issues of particular importance/severity in the LCB?).

- The abstract highlights as one of the core findings that agricultural actors are less likely to participate in forums, but I was not able to find explanation of the analysis linked to this finding.

Minor comments:

- The two ERGM models are similar enough that the authors may consider combining them into one table to make it easier to compare results between the two models.

- In supplemental materials: the GOF plots include diagnostics that seem more appropriate for a “one mode” ERGM than for a bipartite ERGM (e.g., one measure for Degree, rather than Actor Degree and Action Situation Degree; Edge-wise Shared Partners).

- In the first paragraph of “Analysis” (2.2), some clarifications could improve interpretation – for example, in the phrase “…many actors having few edge connections to other nodes”, does “other nodes” refer to action situations? If so, the authors may consider stating that. The phrase “The overall degree distribution of the network is exponential” is a little bit confusing because, to my understanding, the network being discussed is bipartite, with a degree distribution for actors and for forums (and the overall degree distribution combines the two, which complicates interpretation).

- The authors may consider adding a short explanation of the rationale for estimating the first ERGM (i.e., rather than just estimating the second).

Reviewer #2: The paper is generally well-written and easy to follow. I also like how it attempted to connect and apply the lenses of collaborative governance/policy networks (the ecology of games literature) and adjacent/linked action situations (a new spin in the IAD framework and the Bloomington school of institutional analysis) in analyzing the governance of a social-ecological system.

But there is one major issue. It is not clear to me what research gap is addressed by this paper and what is its contribution to the literature. After providing big picture/general knowledge in much of the intro section, the authors suddenly provide a set of expectations/hypotheses in the last paragraph of that section (lines 106-117). I do not know where those come from and why those are important because there is no “theoretically” detailed, thorough, and convincing story leading to those expectations in the initial part of the paper. What are key theoretical puzzles that remain unresolved that you want to answer with regards to the impacts of scale, issues of concern, and scale issue homophily, and joint collaboration on actor participation in the action situations? What are the findings of existing studies, gaps/puzzles that remain, and how does your research address them? Because of these are unclear, the paper was unsatisfying to read (although I still liked the work). Certain parts of later sections touch on previous studies (e.g., lines 296-301), but they came too late and felt to be shallow. As a result of this issue, the paper’s discussion and conclusion sections are mostly speculative and lacking in substance (i.e., no punchlines).

My suggestion is that the authors provide a concentrated dose of theoretical motivation directly leading to the expectations/hypotheses in the intro section (not just one paragraph but multiple paragraphs). Then, enrich and improve the discussion and conclusion sections based on that.

6. PLOS authors have the option to publish the peer review history of their article (what does this mean?). If published, this will include your full peer review and any attached files.

Reviewer #1: No

Reviewer #2: No

---

## [Author Response · Author response to Decision Letter 0]

3 Feb 2023

These comments are also included in the uploaded Response to Reviewers

Academic Editor:

ACADEMIC EDITOR: Thank you for submitting your work to Plos One. Two reviewers have now completed their assessments, and, on the basis of these assessments, I am recommending minor revisions. Both reviewers provide constructive criticisms that do not require new analyses, though additional analysis and clarification may be required in some instances. A area for improvement identified by both reviewers concerns the the generality of the results and how these contribute to a body of theory. Revier1 states: `` I believe the paper would benefit from a stronger explanation of the study’s findings beyond the Lake Champlain Basin study setting." Reviewer 2 notes: ``What are key theoretical puzzles that remain unresolved that you want to answer with regards to the impacts of scale, issues of concern, and scale issue homophily, and joint collaboration on actor participation in the action situations? What are the findings of existing studies, gaps/puzzles that remain, and how does your research address them?" Addressing these critiques will, I believe, improve the reach of the contribution and contribute to the scientific soundness of the manuscript for publication.

Response to Editor:

We appreciate the comments of the reviewers, which we believe have contributed to an improved manuscript. Below, we address each comment individually. In particular, we have added more to the literature review, as well as explanation regarding the paper’s contributions to the body of theory and have cleaned up some minor issues regarding model specification and diagnostics.

Reviewer 1

COMMENT R1-1

My main comment: I believe the paper would benefit from a stronger explanation of the study’s findings beyond the Lake Champlain Basin study setting. I address this point in more detail below. My other comments are divided into major and minor points.

REPONSE TO R1-1

We appreciate the suggestion and have added text setting up the contribution in the introduction and have further explicated the contribution in the discussion section. Namely, we discuss the importance of considering context-dependent factors (e.g., scale, homophily) in institutional design aimed at improving participation in collaborative governance forums. We have also elaborated more on how our findings “…extend polycentric governance frameworks that chart how actor participation can connect forums across space and scale by showing how those connections are differentiated by the type of issue and by geographic scale.” At the request of Reviewer 2, we have also added text in the introduction better situating our work within the relevant literature.

COMMENT R1-2

Nearly all of discussion focuses on the specific context of water quality governance in the Lake Champlain Basin. I encourage the authors to highlight the broader implications of the study. Likewise, the authors may consider explaining—more explicitly—the importance of the study setting in the introduction (e.g., is this a model system for studying institutional complexity? Are HAB or other water quality issues of particular importance/severity in the LCB?).

RESPONSE TO R1-2

The LCB’s issues and the TMDL are not unique – there are hundreds of polluted watersheds across the US. The utility of studying the LCB is in the science-based approach to collaborative water governance taken by the Vermont Agency of Natural Resources (and others). It provides a sort of natural experiment in the governance principles central to this paper. We have added some brief text in the introduction to make this clearer. 

COMMENT R1-3

The abstract highlights as one of the core findings that agricultural actors are less likely to participate in forums, but I was not able to find explanation of the analysis linked to this finding.

REPONSE TO R1-3

This was a miswording. It is not about agricultural actors’ participation, but rather actors’ participation in agricultural forums. The abstract has been corrected.

COMMENT R1-4

The two ERGM models are similar enough that the authors may consider combining them into one table to make it easier to compare results between the two models.

RESPONSE TO R1-4

The two results have been combined in what is now Table 4.

COMMENT R1-5

- In supplemental materials: the GOF plots include diagnostics that seem more appropriate for a “one mode” ERGM than for a bipartite ERGM (e.g., one measure for Degree, rather than Actor Degree and Action Situation Degree; Edge-wise Shared Partners).

RESPONSE TO R1-5

We appreciate you catching our mistake. This was a result of an error in a custom “helper” function I wrote to automate some of diagnostic functions (it was printing the incorrect graph). The models have not changed, but now properly report their fit in the supplemental information. We have also added some text in the methods section to elaborate on the gwb2degree term.

COMMENT R1-6

- In the first paragraph of “Analysis” (2.2), some clarifications could improve interpretation – for example, in the phrase “…many actors having few edge connections to other nodes”, does “other nodes” refer to action situations? If so, the authors may consider stating that. The phrase “The overall degree distribution of the network is exponential” is a little bit confusing because, to my understanding, the network being discussed is bipartite, with a degree distribution for actors and for forums (and the overall degree distribution combines the two, which complicates interpretation).

RESPONSE TO R1-6

The first paragraph on the Analysis section is describing the full network, visualized in Figure 1. The degree comment is for all nodes in this network, and the exponential comment also refers to the full graph and is supported by the citation. We later transform the full graph into the bipartite, which is described later in the analysis section when the ERGMs are introduced. We have added a bit of clarifying text in the first paragraph of the Analysis section, which we believe is sufficient.

COMMENT R1-7

- The authors may consider adding a short explanation of the rationale for estimating the first ERGM (i.e., rather than just estimating the second).

RESPOSE TO R1-7

We have added a short preamble to the first ERGM description in the methods section calling back to our research questions and setting up the baseline model. 

Reviewer 2

COMMENT R2-1

The paper is generally well-written and easy to follow. I also like how it attempted to connect and apply the lenses of collaborative governance/policy networks (the ecology of games literature) and adjacent/linked action situations (a new spin in the IAD framework and the Bloomington school of institutional analysis) in analyzing the governance of a social-ecological system.

RESPONSE TO R2-1

Thank you for your kind comments.

COMMENT R2-2

But there is one major issue. It is not clear to me what research gap is addressed by this paper and what is its contribution to the literature. After providing big picture/general knowledge in much of the intro section, the authors suddenly provide a set of expectations/hypotheses in the last paragraph of that section (lines 106-117). I do not know where those come from and why those are important because there is no “theoretically” detailed, thorough, and convincing story leading to those expectations in the initial part of the paper. What are key theoretical puzzles that remain unresolved that you want to answer with regards to the impacts of scale, issues of concern, and scale issue homophily, and joint collaboration on actor participation in the action situations? What are the findings of existing studies, gaps/puzzles that remain, and how does your research address them? Because of these are unclear, the paper was unsatisfying to read (although I still liked the work). Certain parts of later sections touch on previous studies (e.g., lines 296-301), but they came too late and felt to be shallow. As a result of this issue, the paper’s discussion and conclusion sections are mostly speculative and lacking in substance (i.e., no punchlines).

My suggestion is that the authors provide a concentrated dose of theoretical motivation directly leading to the expectations/hypotheses in the intro section (not just one paragraph but multiple paragraphs). Then, enrich and improve the discussion and conclusion sections based on that.

RESPONSE TO R2-2

We have added the requested paragraphs to the introduction section. We now more thoroughly review the literature on the determinants of collaboration. In particular, we explicate the effects of transaction costs, the importance of building trust, the effects of scale and issue/belief homophily, and take a much deeper dive into papers that explore the joint collaboration/collaborative closure dynamic. We have intentionally kept it brief, as the paper is quite long as it is. But it does add depth to the manuscript, including providing weight to our discussion and conclusion sections, and we appreciate the suggestion.

---

## [Editor Report · Decision Letter 1]

23 Feb 2023

Engagement in Water Governance Action Situations in the Lake Champlain Basin

PONE-D-22-24459R1

Dear Dr. Bitterman,

We’re pleased to inform you that your manuscript has been judged scientifically suitable for publication and will be formally accepted for publication once it meets all outstanding technical requirements.

Kind regards,

Jacob Freeman

Academic Editor

PLOS ONE

Additional Editor Comments (optional):

Thanks for submitting your paper to Plos One and supporting open science. The paper reports a nice contribution.
---

## [Editor Report · Acceptance letter]

9 Mar 2023

PONE-D-22-24459R1 

Engagement in Water Governance Action Situations in the Lake Champlain Basin 

Dear Dr. Bitterman:

I'm pleased to inform you that your manuscript has been deemed suitable for publication in PLOS ONE. Congratulations! Your manuscript is now with our production department. 

Kind regards, 

on behalf of

Dr. Jacob Freeman 

Academic Editor

PLOS ONE